# Proteins Do Not Replicate, They Precipitate: Phase Transition and Loss of Function Toxicity in Amyloid Pathologies

**DOI:** 10.3390/biology11040535

**Published:** 2022-03-30

**Authors:** Kariem Ezzat, Andrea Sturchio, Alberto J. Espay

**Affiliations:** 1Clinical Research Center, Department of Laboratory Medicine, Karolinska Institutet, 141 57 Stockholm, Sweden; 2Department of Clinical Neuroscience, Neuro Svenningsson, Karolinska Institutet, 171 76 Stockholm, Sweden; andrea.sturchio@ki.se; 3James J. and Joan A. Gardner Family Center for Parkinson’s Disease and Movement Disorders, Department of Neurology, University of Cincinnati, Cincinnati, OH 45221, USA; espayaj@ucmail.uc.edu

**Keywords:** amyloid, prion, Alzheimer’s, Parkinson’s, Creutzfeldt–Jakob disease (CJD)

## Abstract

**Simple Summary:**

Amyloid aggregation of proteins in disease has been known for over a hundred years; however, effective therapeutics for amyloid pathologies such as Alzheimer’s disease and Parkinson’s disease are still lacking. This review divides the amyloid phenomenon into four major questions: *What are amyloids? How do amyloids form? Can proteins replicate? How do amyloids cause toxicity?* The aim is to answer these questions within a unified physicochemical framework that links the structural biology of amyloids to the thermodynamics of amyloid formation and the pathophysiology of amyloid aggregates in different diseases. We illustrate that the thermodynamics of protein aggregation does not support the prion protein-only replication hypothesis, and how the structural biology of amyloids makes them largely domainless, generic, and inert. The implications of this understanding for the etiology, pathogenesis and potential therapeutics of amyloid diseases are briefly discussed.

**Abstract:**

Protein aggregation into amyloid fibrils affects many proteins in a variety of diseases, including neurodegenerative disorders, diabetes, and cancer. Physicochemically, amyloid formation is a phase transition process, where soluble proteins are transformed into solid fibrils with the characteristic cross-β conformation responsible for their fibrillar morphology. This phase transition proceeds via an initial, rate-limiting nucleation step followed by rapid growth. Several well-defined nucleation pathways exist, including homogenous nucleation (HON), which proceeds spontaneously; heterogeneous nucleation (HEN), which is catalyzed by surfaces; and seeding via preformed nuclei. It has been hypothesized that amyloid aggregation represents a protein-only (nucleic-acid free) replication mechanism that involves transmission of structural information via conformational templating (the prion hypothesis). While the prion hypothesis still lacks mechanistic support, it is also incompatible with the fact that proteins can be induced to form amyloids in the absence of a proteinaceous species acting as a conformational template as in the case of HEN, which can be induced by lipid membranes (including viral envelopes) or polysaccharides. Additionally, while amyloids can be formed from any protein sequence and via different nucleation pathways, they invariably adopt the universal cross-β conformation; suggesting that such conformational change is a spontaneous folding event that is thermodynamically favorable under the conditions of supersaturation and phase transition and not a templated replication process. Finally, as the high stability of amyloids renders them relatively inert, toxicity in some amyloid pathologies might be more dependent on the loss of function from protein sequestration in the amyloid state rather than direct toxicity from the amyloid plaques themselves.

## 1. What Are Amyloids?

Proteins, like any other molecules, can exist in different states or phases depending on their packing density. Similar to gas, liquid, and solid phases of water, for example, (water vapor, liquid water, and ice), proteins can be soluble or colloidally dispersed in the aqueous biological environment, concentrated in liquid droplets that form a separate liquid phase within the aqueous environment, or in a tightly packed solid state. The liquid–liquid phase separation of proteins has been intensively investigated and reviewed recently [1,2]. Here, we focus on amyloids as specific form of protein solids.

There are two main types of protein solids that form in-vivo:Fibrous proteins, such as actin, elastin, and collagen;Amyloids, which are associated with many human diseases.

While both types of in vivo protein solids share similar physicochemical mechanisms of formation (nucleation and growth, see below), they differ in many fundamental ways. Fibrous proteins, such as actin or collagen fibers, are formed from a specific group of proteins, where the monomers (for examples G-actin and tropocollagen for actin and collagen fibers, respectively) are natively folded before assembling into fibers in a controlled and reversible manner, which involves energy-dependent processes (enzymes and ATP) [3,4,5]. In contrast, amyloid fibrils can be formed by almost any amino acid sequence, from globular nonfibrous proteins, such as myoglobin [6], insulin [7], and albumin, [8] to simple polylysines, polyglutamates, and polythreonines sequences [9]. Such generic nature- and sequence-independence indicate that the architecture of the amyloid state is not encoded in the primary sequence of proteins [10,11,12,13]. Unlike native protein folding, which depends on specific *intramolecular* interactions between the *side chains* of a particular sequence, the structure of the amyloid state is dominated by *intermolecular* interactions via the *backbone* that is common to all proteins. Consequently, amyloids from different proteins possess a common core conformation, the cross-β conformation, where intermolecular β-sheets pair tightly together with their side chains interdigitating (like zipper teeth) excluding water to form the so-called “dry steric zipper” (Figure 1) [14,15,16,17,18]. The intermolecular β-sheets form via a generic interbackbone hydrogen-bonding network between the amide N-H and C=O groups of adjacent protein molecules [19] and can comprise up to thousands of molecules extending for µm distances [14]. Within the β-sheet ladder, β strands are spaced 4.8 A° and the distance between opposite β sheets are in the range of 6–12 A° (Figure 1), which gives rise to the characteristic amyloid X-ray diffraction pattern with meridional and equatorial reflections of similar values, respectively [16,18]. Extended ladders of interdigitating β-sheet pairs form the core spine of the superstructural subunit of amyloids, the protofilament. A protofilament can accommodate a single or multiple steric zippers in different arrangements with different mating/interdigitation options between the side chains of the constituting ladders [15]. Protofilaments further associate laterally into fibrils, which further associate and precipitate as insoluble plaques, characteristic of tissues affected by amyloidosis [20].

Another major difference between native protein folding and the cross-β conformation of amyloids is that the interdigitation of side chains between β-sheet ladders generally deprives amyloids from any characteristic domains. Additionally, the extensive, hierarchical self-interaction (ladders within zippers, zippers within protofilaments, protofilaments within fibrils, and fibrils within plaques) makes amyloids very stable and, consequently, extremely difficult to solubilize and relatively inert. The aggregated, plaque nature of amyloids is, again, in contrast with functional fibrous proteins which assemble in well-defined networks by accessory proteins [23]. The difference between functional fibrous proteins and amyloids is summarized in Table 1.

Despite the universal cross-β conformation of any amyloid, the difference in number or arrangement of steric zippers within a protofilament and/or the number and arrangement of protofilaments within a fibril result in different polymorphs. Unlike the cross-β conformation which is structurally encoded in any protein sequence via generic inter-backbone hydrogen-bonding, polymorphism is dependent on environmental factors such as temperature, pH, concentration, and shaking; extrinsic factors that are not structurally encoded [24]. For example, with the same sequence, different polymorphic shapes of Aβ40 fibrils can be produced in quiescent versus agitating conditions [25], and the presence or absence of polyanions leads to the production of different polymorphs of α-synuclein fibrils [26]. The interaction between environmental factors and the protein sequence affects polymorphism by affecting the patterns of β-sheet ladder stacking and zipper interdigitation, which can also lead to different polymorphs of different sequences under same conditions [15].

As proteins have the information to fold natively in the primary sequence of side chains (Anfinsen’s dogma [27,28]), they also holds the necessary information to form the cross-β conformation based intermolecular backbone interactions, which requires molecular proximity. This is why amyloid formation requires supersaturated conditions and the likelihood of a protein forming an amyloid increases with concentration [29,30]. Above a certain concentration, the molecular proximity renders the intermolecular interactions more favorable than intramolecular interactions responsible for native folding, leading to amyloid formation. This intermolecular interaction will result in molecular packing, phase transition, and precipitation out of the aqueous biological environment. Such phase transition is a spontaneous (exergonic) reaction at supersaturated conditions but will require crossing a thermodynamic barrier, the nucleation barrier.

## 2. How Do Amyloids Form?

Amyloid formation is essentially a process of protein crystallization [30,31]. The difference between the formation of protein crystals ex vivo for structural determination and amyloid formation is that amyloids always adopt a single conformation, the cross-β, while protein crystals hold protein monomers in their native conformation. Otherwise, the physicochemical processes underlying protein crystallization and amyloid formation are similar in terms of the thermodynamic and kinetic parameters governing both. According to the second law of thermodynamics and Gibbs free energy equation, the unfavorable decrease in entropy (increase in order) due to the formation of an ordered solid (crystal/amyloid) is overcome by an increase of the entropy of the solvent (water) due to its expulsion out of the crystalline structure or amyloid fibrils [32,33,34]. A similarity can be traced to the processes that underlie hydrophobic effect, which governs many processes in nature, including protein folding [34,35,36]. In addition to this entropic driving force, amyloid formation is exothermic [37,38], which adds an enthalpic component to the driving force for amyloid phase transition. The Gibbs free energy equation of this process is as follow:(1)$$\Delta G_{amyl}=\Delta H_{amyl}-T\left(\Delta S_{protein\;}+\Delta S_{solvent}\right)$$
where $\Delta G_{amyl}$ is the free energy for amyloid formation, $\Delta H_{amyl}$ is the enthalpy for amyloid formation, *T* is the temperature, $\Delta S_{protein\;}$ is the soluble protein entropy, and $\Delta S_{solvent}$ the solvent entropy.

At a certain level of supersaturation, amyloid formation becomes thermodynamically favorable; however, phase transition will not proceed unless a nucleation barrier is overcome. According to the widely accepted classical nucleation theory, the initial formation of a solid phase within a liquid phase requires overcoming the interfacial energy cost of creating a new interface between the solid phase (amyloid in this case) and the liquid (aqueous environment) [39,40]. Overcoming this barrier requires the formation of a nucleus of a certain size (of radius *r*). Below this radius, nuclei will dissociate back to monomers, and only above it, the system will proceed into phase transition into a solid (Figure 2) [41,42]. With the addition of the nucleation barrier, the free energy equation of amyloid formation becomes as follows:(2)$$\Delta G_{amyl}=\Delta H_{amyl}-T\left(\Delta S_{protein\;}+\Delta S_{solvent}\right)+4\pi \;r^{2}\;\sigma$$
where *r* is the radius of the nucleus and σ is the surface tension of the interface between the nucleus and the solvent. While a nucleus of the right size can spontaneously form at very high supersaturations via a process termed homogenous nucleation (HON), in practice, nucleation usually takes place at interfaces via a process termed heterogenous nucleation (HEN), where the surface helps lower the energy barrier to nucleation by acting as a nucleation site, sparing the interfacial energy required to form new interface in the bulk of the fluid (Figure 2). HEN adds a new term to the equation which is a function of the wetting angle (θ) between the protein and the surface, according to the spherical cap approximation model [42,43]. The free energy equation of HEN can be described as follows:$$\Delta G^{HEN}=\Delta H_{amyl}-T\left(\Delta S_{protein\;}+\Delta S_{solvent}\right)+\left(4\pi \;r^{2}\;\sigma \times f\left(\theta \right)\right)$$
where θ is the wetting angle between the protein and the surface and $f\left(\theta \right)=\frac{2-3\;cos\theta +cos^{3\;}\theta \;}{4}$.

Thus, the final equation of HEN becomes:$$\Delta G^{HEN}=\Delta H_{amyl}-T\left(\Delta S_{protein\;}+\Delta S_{solvent}\right)+\left(4\pi \;r^{2}\;\sigma \times \frac{\;2-3\;cos\theta +cos^{3\;}\theta \;}{4}\right)\;$$

The higher the affinity of the protein to the surface, the lower the wetting angle θ, which will lead to more significant reduction of the nucleation barrier. In this regard, numerous interfaces have been shown to induce amyloid nucleation of proteins via HEN. Interfaces such as lipid surfaces [44,45,46,47], nanoparticles [48,49,50], and viruses [51], in addition to polymer surfaces, such as heparin, glycosaminoglycans [52,53], and nucleic acids, [54] have been shown to facilitate amyloid nucleation. The growing fibril surface itself can serve as a site for HEN, in a phenomenon termed secondary nucleation [55]. Additionally, container surfaces and even air bubbles that result from sonication, agitation, or mechanical stress can lead to amyloid formation of pharmaceutical proteins, such as insulin, during production, transportation, or administration, significantly diminishing its activity [56,57]. The wide variety of interfaces that can facilitate amyloid nucleation indicate the high potency of HEN as the major pathway of amyloid nucleation. In addition to the HON and HEN pathways, the introduction of a preformed nucleus will trigger phase transition by eliminating the nucleation barrier altogether, in a process termed *seeding*. However, seeds can also facilitate nucleation via HEN by acting as promiscuous surfaces [58]. Seeding and HEN pathways are demonstrated for the crystallization of a supersaturated solution of sodium acetate in video 1 (https://youtu.be/e82suzAi3sA (accessed on 15 February 2022)).

Nucleation and growth are also essential for the formation of fibrous functional proteins such as actin and collagen; however, in a more controlled manner. To prevent random nucleation, collagen monomers, for example, include propeptides that need to be cleaved by special proteinases to generate tropocollagen monomers, which are then able to nucleate and grow into fibers [5]. In the case of actin, the G-actin monomer is protected by accessory proteins such as profilin, which require enzymatic cleavage before nucleation and assembly into actin fibers (F-actin) [4]. Moreover, several dedicated, enzymatically activated nucleator protein complexes, such as the Arp2/3 complex, spire, and formin, are present to facilitate and control actin nucleation via HEN [59,60]. Formation of higher-order networks, and disassembly and degradation of such fibrous proteins are also well-controlled via accessory proteins and enzymatic processes [23,61,62]. In addition to such energy intensive (ATP) control mechanisms of nucleation and growth, fibrous proteins harbor high proline and glycine content, which reduces the potential of forming amyloid fibrils [5,63,64]. Taken together, while biology made use of the thermodynamics of nucleation and growth to produce functional fibrous proteins, it also evolved structural and energy-dependent mechanisms to control this process and prevent uncontrolled or irreversible amyloid fibril formation. Such control mechanisms are lacking in amyloid aggregation, where the process is solely dominated by thermodynamic forces (Table 1). Thus, after the nucleation barrier has been crossed, amyloid formation becomes spontaneous and irreversible, and will continue until all the available substrate is transformed into plaques, a process similar to crystallization video 1 (https://youtu.be/e82suzAi3sA (accessed on 15 February 2022)).

## 3. Can Proteins Replicate?

Long before the structural properties and kinetics of amyloid formation were understood, amyloids involved in neurodegenerative diseases such as Kuru and Creutzfeldt-Jacob disease were hypothesized to be “proteins that acquire alternative conformations that become self-propagating”, and labelled as proteinaceous infectious particles, or prions [65]. Prions were assumed to encode conformational information, come in different “strains”, and act as corruptive templates that incite a chain-reaction of misfolding and aggregation [66]. However, nearly 40 years after the inception of the prion hypothesis, (also known as the protein-only hypothesis), many fundamental questions regarding the mechanism of replication and drivers of toxicity remain unanswered [67]. Moreover, the current knowledge about the structure and thermodynamics of amyloid formation do not support the initial assumptions of the prion hypothesis.

Proteins hold the necessary information both for their native folding conformation and for the amyloid cross-β conformation in the primary side chain sequence and the backbone, respectively. Neither of these conformational states requires templating. While a protein adopts its native conformation at lower concentration, adopting the cross-β conformation depends on the degree of supersaturation and the availability of nucleating agents and not on the presence of another protein particle that acts a conformational template. This fact is illustrated clearly by the ability of lipid surfaces, nanoparticles, and viruses to catalyze the formation of amyloids from supersaturated protein solutions via HEN in absence of any protein seed template. Supersaturation provides the molecular proximity required for favoring generic intermolecular backbone interactions over specific intramolecular side-chain interactions, and interfaces can offer a surface to nucleate upon, which enables the crossing of the nucleation barrier. Once the nucleation barrier is crossed, amyloid formation proceeds spontaneously, leading to precipitation of the available soluble protein substrate into insoluble plaques. The uncontrolled nature of this process and the accessibility of the cross-β conformation to any sequence or sequence combinations is responsible for polymorphism, which is dependent on the environmental conditions that favor different ladder, steric zipper and protofilament arrangements. Such sensitivity to environmental factors leads to polymorphic heterogeneity and great intra and inter sample variability [24].

Polymorphism is hypothesized to underlie the phenomenon of prion “strains”, where certain protofilament or fibrillar polymorphs can induce the formation of homogenous plaques, composed solely of protofilaments or fibrils of a morphology similar to the morphology of the seeding “template” via elongation at the seed fibril ends. However, no definitive structural evidence for these presumptions has come forward, and the “strainness” of prions is still diagnosed using such tools as differential resistance to disaggregation and proteolysis [67]. Additionally, there is no thermodynamic basis, in terms of energies and driving forces, for a natively folded soluble protein to exit its thermodynamically stable conformation to bind on top of an amyloid fibril or for a specific fibrillar seed to be able to template its morphology onto soluble protein molecules in a repetitive, “self-propagating” manner. As reviewed above, the process of amyloid formation is a spontaneous phase transition under supersaturated conditions after crossing the nucleation barrier. Therefore, a particular seed morphology cannot control nor steer such a spontaneous process. The end result is a multitude of polymorphs that are not structurally encoded in the seed, but instead depend on the microenvironmental conditions [68]. This is further compounded by the fact that seeds and fibrils can induce HEN via their surfaces and not only via monomer addition to their tips, leading to cross seeding or secondary nucleation. In both cases, there is no mechanism by which the seed can restrict growth to the tips instead of HEN on the surface.

*Polymorphs depend on the recipient conditions, not on donor seeds.* Prions are compared to DNA as an alternative way of transmission of biological information. However, the spontaneous (exergonic) nature of amyloid formation is in contrast to the non-spontaneous (endergonic) nature of DNA replication, which involves continuous energy input in the form of deoxynucleoside triphosphate (dNTP) hydrolysis together with extensive and strict enzymatic control of every step of replication. An example to illustrate the difference between endergonic DNA replication and exergonic amyloid phase transition is demonstrated in Figure 3, where an oligonucleotide and a peptide at high concentrations are treated with nucleating surfaces such as nanoparticles or viruses. The oligonucleotide will not be able to replicate the information in its sequence as it requires enzymes and continuous energy supply from dNTPs. However, the peptide can readily precipitate into amyloids via the spontaneous process of phase transition, where the driving force is the free energy difference between the soluble and the solid state under supersaturated conditions (see above). All that is required is crossing the nucleation barrier, which can be achieved by HEN, seeding, or by simply increasing the peptide concentration to facilitate HON. All these pathways will lead to the generic intermolecular cross-β conformation, which *is* the amyloid conformation encoded in the protein backbone structure and does not need to be templated or transferred. Polymorphism on the other hand is dependent on factors in the recipient environment that are not structurally encoded, and, hence, cannot be faithfully replicated. Importantly, in the absence of supersaturation, seeds will not be able to initiate, let alone template, amyloid formation (Figure 3B).

While the cross-β conformation is structurally encoded and accessible to any protein sequence under the right conditions, proteins differ in their relative propensity to form amyloids [10]. This depends on factors such as the enrichment of amino acids with high β-sheet forming propensities, the overall solubility of the protein, its level of supersaturation, and its exposure to nucleating agents. A combination of these factors leads to selective vulnerability of certain proteins, cells, and tissues for amyloid aggregation. Particularly, supersaturation is a major driving force for protein aggregation [29]. By analyzing single-cell transcriptomic and subcellular proteomics data, Freer et al. found that the most supersaturated proteins are enriched in cells and tissues that succumb first to neurodegeneration (Figure 4) [69,70]. They also showed that the supersaturated proteins are closely involved in synaptic processes, resulting in a high vulnerability of the synaptic environment to aberrant protein aggregation, and that the supersaturation signature coincides with the pattern of disease progression in Alzheimer’s disease (AD). This indicates that physicochemical factors of the recipient environment dictate not only polymorphism, but also the vulnerability to amyloid aggregation and the apparent propagation patterns. Additionally, the diversity of local factors that contribute to protein aggregation can explain the varied patterns of cellular and tissue vulnerabilities that result in different pathologies, despite the similar properties of amyloids. Genetic mutations can also aggravate vulnerability by making a protein unstable, less soluble, or overexpressed. This is the case in many pathogenic mutations that cause neurodegenerative diseases. For example, the H50Q mutation in *SNCA*, which codes for α-synuclein, results in a 10-fold decrease in its solubility, which increases α-synuclein supersaturation and, hence, its propensity to aggregate [71]. Gene duplications in *APP*, such as in some cases of familial AD and Down’s syndrome, and in *SNCA*, such as in familial Parkinson’s disease, will also lead to higher supersaturation and lower the barrier for aggregation. With increasing amyloid formation there is consequent protein consumption, ultimately leading to lower levels of soluble Aβ and α-synuclein, as has been observed clinically [72,73].

## 4. How Do Amyloids Cause Toxicity?

Amyloid formation involves three pathological protein transformations: *structural*, from natively folded to the cross-β conformation; *biophysical*, from soluble to insoluble; and *biological*, from functional to non-functional [68]. The cross-β conformation buries the once-functional domains of the protein within the steric zipper architecture, which makes amyloids extremely stable [14], and, consequently, relatively inert. Additionally, the uncontrolled phase transition leads to loss of protein solubility and colloidal stability resulting in precipitation into plaques, which further buries any potential unpaired side chains via hierarchal self-interaction (see above). Within plaques, amyloid protofilaments and fibrils adopt different polymorphic morphologies depending on environmental conditions. However, since all polymorphs are based on the same cross-β conformation, where the functional side chain domains are sequestered, they are expected to have similar, generic amyloid properties in terms of stability, insolubility, and low reactivity. Furthermore, the loss of protein solubility and colloidal stability favors precipitation and cluster formation over propagation of single fibrils, which requires colloidal dispersion. This is supported by clinical findings that plaques (heterogenous fibrillar clusters) are the hallmarks of amyloid pathologies.

With the high stability and low reactivity, amyloids pose little direct toxicity unless they physically remodel a tissue, for example, in the rare cases of systemic amyloidosis such as immunoglobulin light chain amyloidosis and transthyretin amyloidosis [74,75]. This is especially true for amyloid accumulation within the muscle tissue (e.g., cardiac amyloidosis), where amyloid infiltration physically impair muscle contractility [76]. However, in many other cases, amyloids exist as a benign mass similar to other benign masses, such as fibromas and lipomas. In insulin-derived amyloidosis, for example, repeated injection of insulin subcutaneously in the same spot leads to the creation of insulin amyloid lumps in some diabetic patients [77]. Despite the benign nature of such lumps, patients lose the ability to control glucose levels due to insulin sequestration in the form of amyloid aggregates [78]. Patients are, therefore, instructed to change the location of insulin injections to avoid local aggregation. In this case, the toxicity due to amyloid aggregation is due to loss-of-function (LOF) of the injected insulin and not due direct toxicity from the amyloid mass. Pathogenesis due to LOF is also demonstrated in the case of p53 amyloid formation. P53 is a tumor suppressor protein whose dysregulation or inactivation is involved in more than 50% of all cancers [79]. It has been shown that p53 can form amyloid fibrils leading to enhanced cell proliferation due to its sequestration and LOF [80,81]. These findings clearly indicate that amyloids are not necessarily cytotoxic as they can enhance, not impair, cell proliferation via a LOF mechanism of p53. LOF is also the mechanism behind many phenotypes in yeast due to amyloid formation [82]. For example, amyloid formation of Sup35, which is an essential translation termination factor, induces lethality due to LOF as a result of its sequestration in the amyloid state [83]. Such an outcome can be reversed by supplying the yeast with a modified version of Sup35, where the residues more prone to amyloid formation are removed, while the domains involved in translation termination are maintained [84]. This replacement approach to overcome amyloid LOF toxicity is also utilized clinically in the treatment of diabetes mellites by using pramlintide, which is a less aggregating analogue of the peptide hormone amylin, whose amyloid aggregation in the pancreas and depletion in the circulation is common among diabetic patients [85]. The relatively benign nature of amyloid plaques can also be seen in neurodegenerative diseases such as AD, where up 30% of individuals who have plaques in their brains are cognitively normal [86]. We have recently shown that higher levels of soluble Aβ42 are associated with normal cognition and preservation of brain volume among amyloid positive individuals, regardless of and despite increasing levels of brain amyloid, indicating that LOF of the soluble Aβ42 is more detrimental to neurons than direct gain-of- function (GOF) toxicity from plaques [87]. This suggests that a replacement approach might also be feasible for AD treatment and other neurodegenerative diseases, an alternative to continuing with anti-amyloid strategies, which have invariably failed [88].

*“Toxic oligomers”?* Oligomers have been postulated to explain the lack of association between amyloid plaque load and toxicity, especially in AD. The term oligomers denotes low and medium molecular weight aggregates that are assumed to mediate the amyloid toxicity [89]. However, the evidence of their toxicity has been shown in vitro, not clinically, and the clinically relevant toxic oligomer remains unknown [90]. Moreover, under supersaturated conditions, which are necessary for amyloid formation, the distinction between oligomers and nuclei is hard to make, since the formation of any cluster stable enough will trigger phase transition into fibrils, whereas unstable clusters will dissociate back into monomers (see above). This has been demonstrated experimentally, where the majority of oligomers were shown to dissociate into monomers, in good accordance with the classical nucleation theory [91]. Moreover, the reduction of soluble Aβ42 levels during the disease course reduces the substrate for the oligomers, questioning the long-term effect of such species. This is supported by the fact that attempts to quantify oligomeric species of Aβ found that they are less in AD patients compared to controls [92,93,94].

*“Ratios”*. Despite the progressive decrease in the absolute levels of soluble Aβ42 in AD, the ratio of Aβ42 relative to other shorter versions of the peptide, such as Aβ40, was hypothesized to increase the likelihood of Aβ42 forming aggregates due to its more amyloidogenic nature [95]. However, amyloid aggregation is dependent on supersaturation, which is dependent on the absolute concentration of the peptide. Decreasing peptide concentration will decrease, not increase, its propensity to form any type of aggregates irrespective of its relative levels compared to other peptides. Moreover, it has been recently demonstrated that CSF Aβ42/Aβ40 ratio also decreases during the course of AD [72].

Another indication on the importance of LOF as a pathogenic mechanism in neurodegenerative diseases is the fact that animal models where the amyloidogenic proteins were knocked out or down display phenotypes that resemble those obtained by protein overexpression and aggregation. This has been demonstrated for Aβ42 [96,97] in AD, α-synuclein in Parkinson’s disease [98,99,100], and other neurodegeneration-related proteins that form amyloids such as Tau [101], PrP [102], SOD1 [103], and TDP43 [104]. The similarity of phenotype in both the presence and absence of aggregates can only be explained by LOF mechanism, where the sequestration of protein due to aggregation mimics the effects of gene knock down. Additionally, in many cases, the phenotype can be rescued by restoration of normal soluble levels of these proteins [97,99]. For further discussion on LOF toxicity, we refer the reader to our earlier reviews [68,105].

## 5. Conclusions

Amyloid formation is a phase transition process, which leads to the formation of a special type of protein solids where proteins assume the cross-β conformation and precipitate in the form of plaques. Cross-β *is* the amyloid conformation, which is a generic intermolecular conformation based on backbone hydrogen bond interactions. It comprises extended β-sheet ladder pairs with interdigitating sidechains (steric zippers), which renders amyloids insoluble, largely domainless, highly stable, and relatively inert. Polymorphism on the other hand refers to different cross-β associations within protofilaments or fibrils, which is a function of environmental conditions. The information to form cross-β is present in the backbone of any protein sequence and does not require to be templated or transferred, while polymorphism is a function of environmental conditions and cannot be structurally encoded nor transferred. What “corrupts” a protein and results in amyloid formation are supersaturation and nucleation, not templating. Supersaturation provides the molecular proximity that facilitates the formation of generic intermolecular backbone interactions rather than the specific intramolecular side-chain interactions required for native folding. Nucleation can take place via HON, HEN, or seeding, where HEN is the most common nucleation pathway that can be induced by a multitude of surfaces including viral envelopes, in the absence of amyloid seeds/prions. After the nucleation barrier is overcome, amyloid growth into heterogenous plaques is spontaneous and uncontrollable, leading to the consumption of the available soluble protein substrate. This contrasts with the controlled nucleation of functional protein fibers, such as actin and collagen, which involves enzymes and ATP. The spontaneous (exergonic) nature of amyloid aggregation is also in contrast to DNA replication, which is a well-controlled, endergonic process. Finally, while amyloid fibrils are largely inert, toxicity in some amyloid pathologies might be more dependent on soluble protein depletion, as they are sequestered into plaques, a LOF mechanism of toxicity, which opens the door for testing new therapeutics based on protein replacement.

## Figures and Tables

**Figure 1 biology-11-00535-f001:**
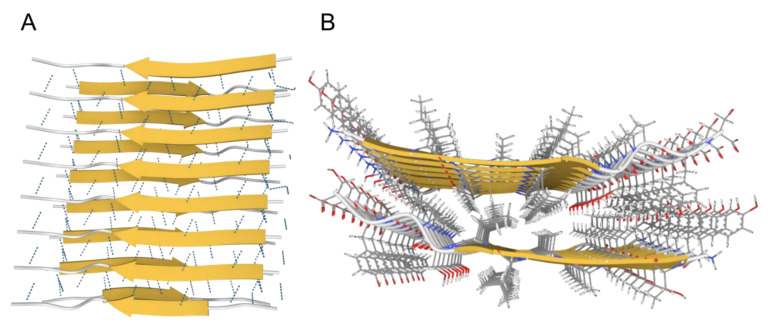
The cross-β conformation of amyloids. (**A**). Side-view showing the intermolecular β-sheets stabilized by hydrogen bonds (dotted lines). (**B**). Top-view showing the dry steric zipper between the two opposing β-sheets with interdigitating sidechains. Images created using Mol* [21] from PDB structure 2M5N from paper by Fitzpatrick et al., 2013 [22].

**Figure 2 biology-11-00535-f002:**
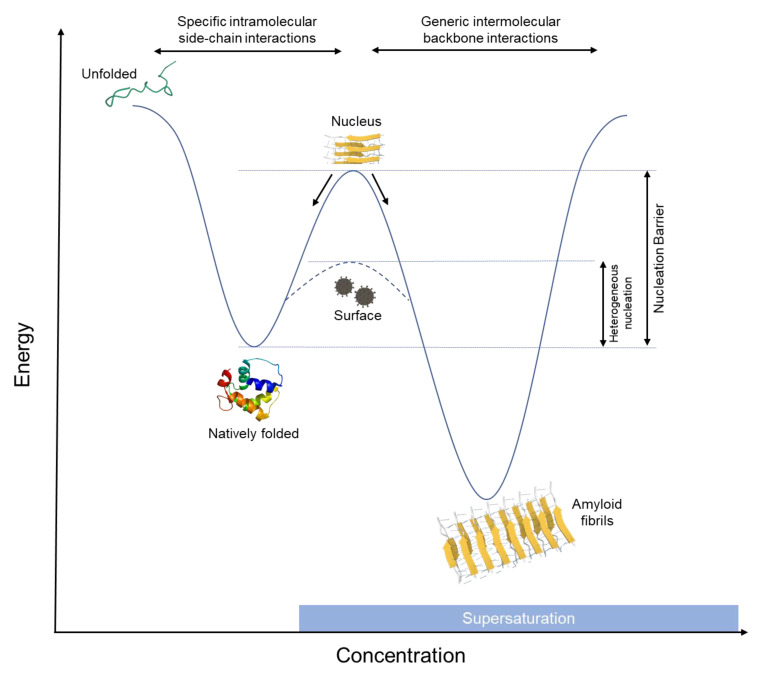
Energy landscape of protein folding. Under unsaturated and saturated conditions, proteins assume their thermodynamically favorable, native conformation based on specific interactions between the sidechains constituting their primary sequence. Under supersaturated conditions, molecular proximity render generic intermolecular backbone interactions more likely; however, phase transition into amyloids will not take place unless a nucleation barrier is crossed. A nucleus of certain size needs to be formed, smaller nuclei dissociate back into monomers, while larger nuclei will trigger phase transition. Nucleation can be triggered by addition of preformed nuclei (seeding) or catalyzed by surfaces, which will lower the energy barrier to nucleation (HEN).

**Figure 3 biology-11-00535-f003:**
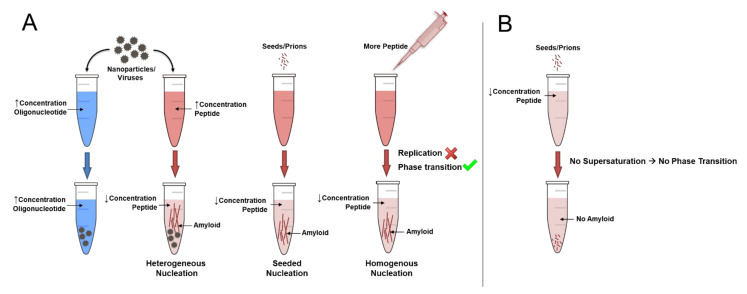
Proteins precipitate, they do not replicate. (**A**). The information to form amyloids is already present in the backbone of any protein sequence. Under supersaturated conditions, any protein can be triggered to precipitate into amyloids by surfaces (HEN), a preformed nucleus (seeding), or increasing the concentration to allow for HON. This is in contrast to DNA replication, which requires a specific template sequence, enzymes and continuous energy input in the form dNTPs. (**B**). Formation of amyloid is dependent on the recipient conditions, not on donor seeds. Thus, adding seeds/prions to proteins in a non-supersaturated condition will not result in amyloid formation, protein unfolding or any kind of templating.

**Figure 4 biology-11-00535-f004:**
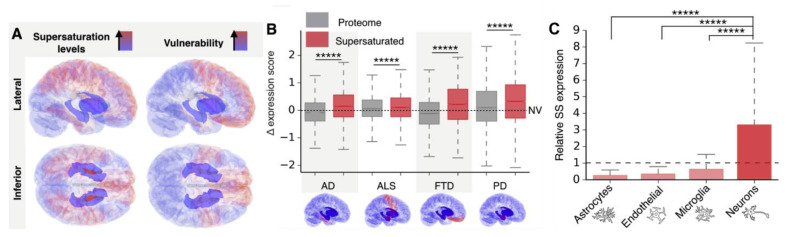
Supersaturation correlates with tissue and cell vulnerability in neurodegenerative diseases (reproduced with permission from Freer et al., 2019 [69]). (**A**). Red regions indicate elevated expression of supersaturated proteins relative to the proteome (left), which are also the tissues most vulnerable to neurodegeneration in the early stages of AD, PD, frontotemporal dementia (FTD), and amyotrophic lateral sclerosis (ALS) (left). (**B**). Scores of the top 5% of supersaturated proteins calculated for AD, PD, ALS, and FTD. (**C**). Relative expression of supersaturated proteins in different cell types. ***** *p* < 0.0005.

**Table 1 biology-11-00535-t001:** The differences between fibrous proteins and amyloid fibrils.

Fibrous Proteins	Amyloid Fibrils
Specific proteins	Any protein sequence
Monomers assemble in their native conformation via specific intramolecular sidechain-based interactions	Proteins assemble into in cross-β conformation via generic intermolecular backbone interactions
Functional domains remain accessible	Majority of functional domains are buried in steric zippers
Form well-defined networks	Precipitate into plaques
Controlled nucleation and growth via structural elements (proline and glycine rich), capping proteins, specific nucleators, enzymes and ATP	Uncontrolled
Reversible	Irreversible

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
