# Peer review of "Proteins Do Not Replicate, They Precipitate: Phase Transition and Loss of Function Toxicity in Amyloid Pathologies"

_biology, 2022, doi:10.3390/biology11040535_

Round 1

Reviewer 1 Report

Protein aggregation /phase transition is a very interesting and hot topic in many diseases models, especially neurodegeneration disease. The authors well explained and discussed the mechanisms behind the Fibrous proteins and Amyloid fibrils formation. They challenged the prion hypothesis through a thermodynamical model and clarified the toxicity in Amyloid pathologies by LOF mechanism. The manuscript was written in good logic and easy to follow. I enjoyed reading this review and it will be helpful for the protein transition community. I only have a few minor format suggestions as follow:

  1. line 250, nucleoside triphosphate (NTP), change to Deoxynucleoside triphosphate (dNTP)
  2. line253, figure 3, change to Figure 3

Author Response

We thank the reviewer for the positive comments. The suggested changes are now included in the manuscript.

Reviewer 2 Report

I have read the Manuscript " Proteins Do Not Replicate, They Precipitate: Phase Transition and Loss of Function Toxicity in Amyloid Pathologies" by Kariem Ezzat, Andrea Sturchio and Alberto J. Espay. The work concerns a discussion on the processes underlying amyloids formation and toxicity as related to corresponding proteins phase transition.

The work is interesting and well written but the aim is not clear. It is submitted as a review manuscript, but a review of what?

Already the abstract misses the aim of the manuscript and the specific topics it will address.

Therefore, in my opinion, the work is an interesting subject material and deserves to be published in Biology but it needs to be reorganized before being accepted.

Author Response

We thank the reviewer for the positive comments. We have now added Simple Summary section before the Abstract where we highlight the aims of the review and the topics it addresses.

Reviewer 3 Report

In this manuscript, the authors pretend to give an interesting review of the importance of heterogeneous nucleation in amyloid aggregation as well as how loss of aggregating protein function is the main cause of the origin of amyloid disease. I think that the underlying idea of this review is very interesting and could be useful to the scientific community if the manuscript was written it with solid and well-defined concepts.

I would only recommend the publication of this manuscript after a considerable number of changes:

  1. In the line 16 the authors claim that the heterogeneous nucleation (HEN) is that is catalyzed by an exogenous surface, but this is not really true since any amyloid fibril surface can to catalyse the heterogeneous nucleation, and this surface type is not exogenous. In fact, it is known like secondary nucleation.

  1. In the line 17 the authors also cite seeding as a nucleation pathways and they claim that the process of seeding is catalyzed by a preformed nucleus. The seeding is not a nucleation pathway; in fact, it is a process that let us to by-pass the nucleation step. Moreover, "catalysed" is not the most appropriate word, since it is not referring to nucleation; this word would be appropriate if the authors were talking about the seed as surfaces that favour secondary nucleation on their surface.

  1. In the line 17 the authors also say “amyloids can be formed from different protein sequences”. Actually, all proteins can give amyloids under the right conditions, not just some sequences [Dobson, C.M. (1999). Protein misfolding, evolution and disease; Dobson, C.M. (2003). Protein Folding and misfolding].

  1. Between lines 49 and 51, I consider that it would be more appropriate to replace reference 10 by the references: Dobson, C.M. (1999). Protein misfolding, evolution and disease; and by Dobson, C.M. (2003). Protein Folding and misfolding. These are the first ones in which C.M. Dobson showed what the authors wish to refer to here with the reference 10.

  1. In Table 1 the authors state that: “Proteins assemble directly into in cross-ß conformation via generic intermolecular backbone interactions”. Perhaps I misunderstood, but they seem to claim that proteins undergo a conformational change that goes directly from their native conformation to the cross-beta structure. The truth is that if the protein has a native fixed in space functional conformation it usually loses it before it aggregates to the amyloid state. Many amyloid proteins are intrinsically disordered proteins, or have intrinsically disordered domains, but even proteins with a well-defined conformation in their native conformational state use to suffer, at least partially, a prior loss of structure.

  1. In Table 1 the authors also claim that the aggregation amyloid process is irreversible, but it is really an equilibrium and it could, in many environment, move towards the monomeric state. In addition, there are certain systems, such as the chaperones, that can act remodelling the amyloid aggregates for give back to monomers.

  1. Between lines 87 and 91, the authors' statement is not true. Polymorphism depends not only on environmental factors but also on the sequence of the protein; the side chains of the main chain are key in the formation of the polymorph. This is why different proteins under identical aggregation conditions can give different polymorphs. Even a mutant of the same protein under the same aggregation conditions can give a different polymorph by altering the side chains of the protein.

  1. Figure 2 is not correct. The decrease in the nucleation energy barrier shows the description "Aberrant surface", but this is not true since both an external surface and the surface of the amyloid fibre itself can catalyse (heterogeneous) nucleation and promote this decrease in nucleation energy. The caption on the right says "Nucleation Barrier", and while it is true that it is the energy barrier of homogeneous nucleation; where it says "Heterogeneous nucleation", it is also a nucleation energy, in this case the nucleation energy of heterogeneous nucleation. Furthermore, in the description of the image it is said that the addition of seeds decreases the nucleation energy, and while this is partially true, since aggregation can be catalysed on their surface, the fact is that the main function of the seeds is by-passing the nucleation stage, since they can directly lead to the elongation stage of the polymorph. Also, there is a typo in the description, there is an extra point at the end.

  1. Between lines 161 and 164 the authors list different surfaces that can catalyse amyloid nucleation and, once again, do not mention the surface of the fibril itself as a surface that can give rise to heterogeneous nucleation.

  1. Between lines 171 and 173 we can finally see the concept of secondary nucleation, which should have been introduced when heterogeneous nucleation was discussed. This information is completely true and contrasts with the statements made in figure 2 or with the idea that aberrant surfaces were the ones that could only give rise to the decrease in the nucleation energy barrier. That is what was understood in the previous sections.

  1. Between lines 218 and 219 the authors claim “foreign interfaces offer a surface to nucleate” but would be better “foreign interfaces can offer a surface to nucleate”. It is so because it is necessary an active surface, not all surfaces can promote the heterogeneous nucleation. It use to require hydrophobic surfaces.

  1. Between lines 241 and 245 what is said about heterogeneous nucleation on the amyloid surface is correct, but it seems to be stated once again that the addition of monomer to the seed ends is a heterogeneous nucleation process. Perhaps it should be rewritten as "seeds and fibrils can induce amyloid aggregation via HEN over their surfaces and not only via monomer addition to their tips". There are many similar sentences that should be rewritten to eliminate this misconception or misexplanation on the part of the authors and that is repeated throughout the text at the different points where I have pointed it out.

  1. I think that the "nucleation barrier" concept that the authors make between lines 259 and 261 is not adequate. The increase in the peptide concentration, the introduction of active surfaces that allow heterogeneous nucleation or other environmental changes will give a decrease in the nucleation energy barrier, which would eventually allow the peptide to overcome it. On the other hand, the addition of seeds to the system bypasses this barrier since the seeds can lead to fibril elongation without the need for a nucleation steep.

  1. It would be advisable to review Figure 3 in light of the previous comments.

  1. The authors should revise the statement they make between lines 324 and 327. During amyloid formation, the fibril in vivo does not have to be part of the cluster from the beginning of its formation, and in fact that is what nucleation is all about. In addition, the amyloid cascade has different species of amyloid nature that can be toxic, as is the case for many oligomeric species. In fact, it has been shown that in many cases it is the oligomeric species that are the most toxic. In addition, the ability of amyloid species to spread the disease from one cell to another is based on the fact that the fibrils can generate fragments that spread through the intercellular medium and thus reach healthy cells, permeating the cellular membrane and promoting seeding inside the healthy cell.

  1. The fragment between lines 365 and 377 could be controversial to say the least, and the authors provide little literature to support their suggestion. This is especially important given that they are questioning the evidence provided by the vast majority of in vitro studies, based on the lack of sufficient clinical studies.

  1. The conclusions follow the general ideas of the rest of the manuscript and maintain the conceptual errors already discussed.

Authors should consider a better review of the references they have chosen, because sometimes they make claims that are not true, and that the references do not in fact support. They should also consider supporting with more references the suggestions they make about the important role in amyloidosis of protein loss of function in the face of amyloid species toxicity.

Author Response

In this manuscript, the authors pretend to give an interesting review of the importance of heterogeneous nucleation in amyloid aggregation as well as how loss of aggregating protein function is the main cause of the origin of amyloid disease. I think that the underlying idea of this review is very interesting and could be useful to the scientific community if the manuscript was written it with solid and well-defined concepts.

I would only recommend the publication of this manuscript after a considerable number of changes:

1. In the line 16 the authors claim that the heterogeneous nucleation (HEN) is that is catalysed by an exogenous surface, but this is not really true since any amyloid fibril surface can to catalyse the heterogeneous nucleation, and this surface type is not exogenous. In fact, it is known like secondary nucleation.

We agree with the reviewer. We have removed the word “exogenous”, and the sentence now reads: heterogeneous nucleation (HEN), which is catalyzed by surfaces

2. In the line 17 the authors also cite seeding as a nucleation pathway and they claim that the process of seeding is catalyzed by a preformed nucleus. The seeding is not a nucleation pathway; in fact, it is a process that let us to by-pass the nucleation step. Moreover, "catalysed" is not the most appropriate word, since it is not referring to nucleation; this word would be appropriate if the authors were talking about the seed as surfaces that favour secondary nucleation on their surface.

We agree with the reviewer.  We have removed the word catalyzed, and the sentence now reads: and seeding via preformed nuclei.

3. In the line 17 the authors also say, “amyloids can be formed from different protein sequences”. Actually, all proteins can give amyloids under the right conditions, not just some sequences [Dobson, C.M. (1999). Protein misfolding, evolution and disease; Dobson, C.M. (2003). Protein Folding and misfolding].

We agree with the reviewer. We have replaced the word “different” by the word “any”, and the sentence now reads: amyloids can be formed from any protein sequence.

4. Between lines 49 and 51, I consider that it would be more appropriate to replace reference 10 by the references: Dobson, C.M. (1999). Protein misfolding, evolution and disease; and by Dobson, C.M. (2003). Protein Folding and misfolding. These are the first ones in which C.M. Dobson showed what the authors wish to refer to here with the reference 10.

We agree with the reviewer, and we have now added the suggested references.

5. In Table 1 the authors state that: “Proteins assemble directly into in cross-ß conformation via generic intermolecular backbone interactions”. Perhaps I misunderstood, but they seem to claim that proteins undergo a conformational change that goes directly from their native conformation to the cross-beta structure. The truth is that if the protein has a native fixed in space functional conformation it usually loses it before it aggregates to the amyloid state. Many amyloid proteins are intrinsically disordered proteins, or have intrinsically disordered domains, but even proteins with a well-defined conformation in their native conformational state use to suffer, at least partially, a prior loss of structure.

We thank the reviewer for raising this point and we now removed the word “directly”.

6. In Table 1 the authors also claim that the aggregation amyloid process is irreversible, but it is really an equilibrium, and it could, in many environments, move towards the monomeric state. In addition, there are certain systems, such as the chaperones, that can act remodelling the amyloid aggregates for give back to monomers.

We agree with the reviewer that the process can be reversible until the nucleation barrier is crossed. However, beyond the nucleation barrier, the process becomes thermodynamically favorable and spontaneous, and thus, cannot be reversed unless external energy is invested. That is the reason behind the extreme stability of amyloids, which makes them extremely resistant to solubilization in detergents for example, and they require harsh conditions (99% formic acid) to revert back into monomers.   

7. Between lines 87 and 91, the authors' statement is not true. Polymorphism depends not only on environmental factors but also on the sequence of the protein; the side chains of the main chain are key in the formation of the polymorph. This is why different proteins under identical aggregation conditions can give different polymorphs. Even a mutant of the same protein under the same aggregation conditions can give a different polymorph by altering the side chains of the protein.

We agree with the reviewer, and we specified this part as follows, “For example, with the same sequence, different polymorphic shapes of Aβ40 fibrils can be produced in quiescent versus agitating conditions [23], and the presence or absence of polyanions leads to the production of different polymorphs of α-synuclein fibrils [24].

8. Figure 2 is not correct. The decrease in the nucleation energy barrier shows the description "Aberrant surface", but this is not true since both an external surface and the surface of the amyloid fibre itself can catalyse (heterogeneous) nucleation and promote this decrease in nucleation energy. The caption on the right says, "Nucleation Barrier", and while it is true that it is the energy barrier of homogeneous nucleation; where it says, "Heterogeneous nucleation", it is also a nucleation energy, in this case the nucleation energy of heterogeneous nucleation. Furthermore, in the description of the image it is said that the addition of seeds decreases the nucleation energy, and while this is partially true, since aggregation can be catalysed on their surface, the fact is that the main function of the seeds is by-passing the nucleation stage, since they can directly lead to the elongation stage of the polymorph. Also, there is a typo in the description, there is an extra point at the end.

We have now corrected the figure by removing the word “aberrant” and the figure legend now reads: Nucleation can be triggered by addition of preformed nuclei (seeding) or catalyzed by surfaces, which will lower the energy barrier to nucleation (HEN).

9. Between lines 161 and 164 the authors list different surfaces that can catalyse amyloid nucleation and, once again, do not mention the surface of the fibril itself as a surface that can give rise to heterogeneous nucleation.

We agree with the reviewer, and we have now moved the mention of secondary nucleation from line 171 up to line 165.

10. Between lines 171 and 173 we can finally see the concept of secondary nucleation, which should have been introduced when heterogeneous nucleation was discussed. This information is completely true and contrasts with the statements made in figure 2 or with the idea that aberrant surfaces were the ones that could only give rise to the decrease in the nucleation energy barrier. That is what was understood in the previous sections.

Figure 2. was modified accordingly.

11. Between lines 218 and 219 the authors claim, “foreign interfaces offer a surface to nucleate” but would be better “foreign interfaces can offer a surface to nucleate”. It is so because it is necessary an active surface, not all surfaces can promote the heterogeneous nucleation. It use to require hydrophobic surfaces.

We agree with the reviewer and the sentence now has been changed accordingly.

12. Between lines 241 and 245 what is said about heterogeneous nucleation on the amyloid surface is correct, but it seems to be stated once again that the addition of monomer to the seed ends is a heterogeneous nucleation process. Perhaps it should be rewritten as "seeds and fibrils can induce amyloid aggregation via HEN over their surfaces and not only via monomer addition to their tips". There are many similar sentences that should be rewritten to eliminate this misconception or misexplanation on the part of the authors and that is repeated throughout the text at the different points where I have pointed it out.

We agree with the reviewer and the sentences have been modified accordingly.

13. I think that the "nucleation barrier" concept that the authors make between lines 259 and 261 is not adequate. The increase in the peptide concentration, the introduction of active surfaces that allow heterogeneous nucleation or other environmental changes will give a decrease in the nucleation energy barrier, which would eventually allow the peptide to overcome it. On the other hand, the addition of seeds to the system bypasses this barrier since the seeds can lead to fibril elongation without the need for a nucleation step.

We fully agree with the reviewer, and we have rewritten this sentence to make it clearer as follows: All what is required is crossing the nucleation barrier, which can be achieved by HEN, seeding, or by simply increasing the peptide concentration to facilitate HON.  

14. It would be advisable to review Figure 3 in light of the previous comments.

The legend of figure 3 has been modified accordingly.

15. The authors should revise the statement they make between lines 324 and 327. During amyloid formation, the fibril in vivo does not have to be part of the cluster from the beginning of its formation, and in fact that is what nucleation is all about. In addition, the amyloid cascade has different species of amyloid nature that can be toxic, as is the case for many oligomeric species. In fact, it has been shown that in many cases it is the oligomeric species that are the most toxic. In addition, the ability of amyloid species to spread the disease from one cell to another is based on the fact that the fibrils can generate fragments that spread through the intercellular medium and thus reach healthy cells, permeating the cellular membrane and promoting seeding inside the healthy cell.

We argue that amyloid fibril formation and the loss of protein solubility will lead to colloidal instability, which will favor precipitation and cluster formation over propagation of single fibrils, which requires colloidal dispersion. This is supported by clinical findings that plaques (heterogenous fibrillar clusters) are the hallmark of amyloid pathologies and not singular fibrillar species. We have now added the loss of colloidal stability explanation to this section.  

16. The fragment between lines 365 and 377 could be controversial to say the least, and the authors provide little literature to support their suggestion. This is especially important given that they are questioning the evidence provided by the vast majority of in vitro studies, based on the lack of sufficient clinical studies.

We have now added more literature support to this section.

17. The conclusions follow the general ideas of the rest of the manuscript and maintain the conceptual errors already discussed. Authors should consider a better review of the references they have chosen because sometimes they make claims that are not true, and that the references do not in fact support. They should also consider supporting with more references the suggestions they make about the important role in amyloidosis of protein loss of function in the face of amyloid species toxicity.

We have agreed to the majority of the reviewer comments and modified the manuscript accordingly. We have also added more references to support our arguments.

Round 2

Reviewer 2 Report

The review is now in a publishable form.

Author Response

The review is now in a publishable form.

We thank the reviewer for accepting the manuscript for publication. 

Reviewer 3 Report

After the corrections made by the authors, the paper is much better, but there are still a few more corrections to be made, which the authors have not done adequately. I believe that in this paper the basic problem is that the authors are discussing a subject on which they are not sufficiently expert, which leads them to not really understand the errors that were pointed out to them in some points. Particularly noteworthy are the conceptual errors in the steps of initiation of amyloid aggregation by seeds.

The following changes are still pending:

  1. Between lines 21 and 24 the authors are remove the word “catalysed” but they keep insisting that seeding is a nucleation pathway when it is not. It is in fact a way of by-pass the nucleation step and thus avoiding the bottleneck it implies. I explained this in the previous report.

  1. I thank the authors for their clarification of point 6 of my report. Their explanation seems to me to be correct and I have nothing more to add.

  1. The authors have not understood point 7 of my previous report. In it I state: "Between lines 87 and 91, the authors' statement is not true. Polymorphism depends not only on environmental factors but also on the sequence of the protein; the side chains of the main chain are key in the formation of the polymorph. This is why different proteins under identical aggregation conditions can give different polymorphs. Even a mutant of the same protein under the same aggregation conditions can give a different polymorph by altering the side chains of the protein". They have answered me by saying that indeed the same sequences can give different polymorphs in different conditions, and that is true. The problem is that under the same conditions different sequences do not give the same polymorphs, which is what I was telling them. Even if two sequences are almost identical (except for a single amino acid, as happens with mutants), they can give different polymorphs in the same aggregation environment. This is because the polymorphism also depends on the side chains of the protein or peptide. The authors deny the latter and have not corrected it in the new version. This is a point that has been widely proven and that has been collected in different scientific papers that the authors themselves are using as citations in this review.

  1. Looking at point 8 of my previous report, I would recommend the authors to heed my suggestion in which I say: “The caption on the right says, "Nucleation Barrier", and while it is true that it is the energy barrier of homogeneous nucleation; where it says, "Heterogeneous nucleation", it is also a nucleation energy, in this case the nucleation energy of heterogeneous nucleation". The problem is that both are energy barriers, which could create confusion in the neophyte reader when viewing the figure. They also did not understand my comment: "in the description of the image it is said that the addition of seeds decreases the nucleation energy, and while this is partially true, since aggregation can be catalysed on their surface, the fact is that the main function of the seeds is by-passing the nucleation stage, since they can directly lead to the elongation stage of the polymorph"; which makes it still wrong. As I have already repeated several times, seed addition is not a pathway of nucleation. Its main purpose is to by-pass the nucleation stage and take the reaction directly to the elongation stage. Both from the review itself and from the authors' response it seems quite clear that they do not possess a deep enough understanding of the early stages of the amyloid aggregation process from a biophysical point of view.

  1. In the point 14 I suggested that the legend of Figure 3 should to be modified in light of the previous comments. Their answer have been: “The legend of figure 3 has been modified accordingly”, but there is no modification. It is as it was at the beginning.

  1. Regarding point 15, as I said in the previous report, it is somewhat controversial and I don't know to what extent the authors are right. I personally believe they are not, and the vast majority of in vitro studies conducted to date give results contrary to the authors' hypothesis. But I understand that there are researchers who disagree, and I think it is healthy for the scientific community to publish their ideas, so I will not insist more than is strictly necessary. But I do think it is necessary to remove the statement: “as heterogeneous clusters and not as separate protofilaments or fibrils that can mediate specific toxicity mechanisms”. As I said in the previous report, this is not true and it has been amply proven in animal model studies that fibrils or fragments of fibrils are transmitted between cells and promote amyloid aggregation in healthy cells. So that statement is not true. Furthermore, as I commented in the previous report, given the controversial claim that amyloidosis is due exclusively to loss of protein function, I think the authors should provide a small literature review of the state of the art to support their claim. However, they hardly provide any bibliography to support it, and if one looks for almost everything that can be found contradicts precisely their proposal. In fact, the physiological function of some amyloidogenic proteins, such as alpha-synuclein (whose aggregation in the form of Lewy bodies is the hallmark of Parkinson's disease), is not even well understood. So it is difficult to argue that it is their loss of function that gives rise to amyloidosis and not the toxic effect of any of the species involved in the amyloid cascade. I repeat that I would find this review very interesting, providing the state of the art in this direction, if a bibliographical study were to be carried out to support this hypothesis, instead of stating it almost without support.

  1. Point 16 went deeper into the problem that had already been advanced in point 15 of the previous report, namely the lack of bibliography to support the authors' hypothesis. The authors' correction has been to add one line and three references, which is far from what would be considered necessary to accept their proposal with a certain degree of rigour. The authors propose a controversial hypothesis for the whole of amyloidosis and all possible amyloid fibrils, and provide hardly any literature and only for amyloid beta peptide. For the rest, I refer to what was said in the previous point. A hypothesis that goes against most of the experimental studies that have been carried out to date is a very interesting topic but one that requires a deep and well-argued discussion and with a multitude of studies to support it. This paper affirms this hypothesis without either of these two requirements.

Overall the article is better, but there are still misconceptions regarding the role of seeds in amyloid aggregation. Once those errors in the seeds role are corrected, the paper would be fine for the introductory part. Another thing is the issue of the role of loss of protein function in the origin of amyloidosis. I think that the authors have the opportunity to make a very interesting review if they develop it well, but now it is not. It is not a question that they have made any error that can be corrected, as has happened with the initiation of amyloid aggregation, but that they have hardly provided data to support their hypothesis. I think that the study they have done is far from being enough since they have focused on Alzheimer's disease without giving more examples, despite the fact that they propose a general mechanism. And within Alzheimer's disease they are based more on the lack of clinical studies than on actual studies that support their hypothesis. I am convinced that an excellent and interesting review can be made if the authors go deeper into this point.

Author Response

After the corrections made by the authors, the paper is much better, but there are still a few more corrections to be made, which the authors have not done adequately. I believe that in this paper the basic problem is that the authors are discussing a subject on which they are not sufficiently expert, which leads them to not really understand the errors that were pointed out to them in some points. Particularly noteworthy are the conceptual errors in the steps of initiation of amyloid aggregation by seeds.

The following changes are still pending:

1. Between lines 21 and 24 the authors are remove the word “catalysed” but they keep insisting that seeding is a nucleation pathway when it is not. It is in fact a way of by-pass the nucleation step and thus avoiding the bottleneck it implies. I explained this in the previous report.

We believe this is a semantic and not a conceptual disagreement. We consider seeding as a pathway of nucleating a system into phase transition, and hence, a nucleation pathway. Meanwhile, we discuss the thermodynamics of nucleation in great detail the section titled “How do amyloids form?”, where we explicitly say, “In addition to the HON and HEN pathways, the introduction of a preformed nucleus will trigger phase transition by eliminating the nucleation barrier altogether, in a process termed seeding. However, seeds can also facilitate nucleation via HEN by acting as promiscuous surfaces [58]”

2. I thank the authors for their clarification of point 6 of my report. Their explanation seems to me to be correct and I have nothing more to add.

We are delighted that the reviewer found our answer satisfactory.

3. The authors have not understood point 7 of my previous report. In it I state: "Between lines 87 and 91, the authors' statement is not true. Polymorphism depends not only on environmental factors but also on the sequence of the protein; the side chains of the main chain are key in the formation of the polymorph. This is why different proteins under identical aggregation conditions can give different polymorphs. Even a mutant of the same protein under the same aggregation conditions can give a different polymorph by altering the side chains of the protein". They have answered me by saying that indeed the same sequences can give different polymorphs in different conditions, and that is true. The problem is that under the same conditions different sequences do not give the same polymorphs, which is what I was telling them. Even if two sequences are almost identical (except for a single amino acid, as happens with mutants), they can give different polymorphs in the same aggregation environment. This is because the polymorphism also depends on the side chains of the protein or peptide. The authors deny the latter and have not corrected it in the new version. This is a point that has been widely proven and that has been collected in different scientific papers that the authors themselves are using as citations in this review.

We agree with the reviewer; and we have now added this sentence with a reference to the end of the paragraph, “The interaction between environmental factors and the protein sequence affects polymorphism by affecting the patterns of β-sheet ladder stacking and zipper interdigitation, which can also lead to different polymorphs of different sequences under same conditions [15].”

4. Looking at point 8 of my previous report, I would recommend the authors to heed my suggestion in which I say: “The caption on the right says, "Nucleation Barrier", and while it is true that it is the energy barrier of homogeneous nucleation; where it says, "Heterogeneous nucleation", it is also a nucleation energy, in this case the nucleation energy of heterogeneous nucleation". The problem is that both are energy barriers, which could create confusion in the neophyte reader when viewing the figure. They also did not understand my comment: "in the description of the image it is said that the addition of seeds decreases the nucleation energy, and while this is partially true, since aggregation can be catalysed on their surface, the fact is that the main function of the seeds is by-passing the nucleation stage, since they can directly lead to the elongation stage of the polymorph"; which makes it still wrong. As I have already repeated several times, seed addition is not a pathway of nucleation. Its main purpose is to by-pass the nucleation stage and take the reaction directly to the elongation stage. Both from the review itself and from the authors' response it seems quite clear that they do not possess a deep enough understanding of the early stages of the amyloid aggregation process from a biophysical point of view.

We refer to our answer on comment 1.

5. In the point 14 I suggested that the legend of Figure 3 should to be modified in light of the previous comments. Their answer have been: “The legend of figure 3 has been modified accordingly”, but there is no modification. It is as it was at the beginning.

The figure legend is amended with track changes.

6. Regarding point 15, as I said in the previous report, it is somewhat controversial and I don't know to what extent the authors are right. I personally believe they are not, and the vast majority of in vitro studies conducted to date give results contrary to the authors' hypothesis. But I understand that there are researchers who disagree, and I think it is healthy for the scientific community to publish their ideas, so I will not insist more than is strictly necessary. But I do think it is necessary to remove the statement: “as heterogeneous clusters and not as separate protofilaments or fibrils that can mediate specific toxicity mechanisms”. As I said in the previous report, this is not true and it has been amply proven in animal model studies that fibrils or fragments of fibrils are transmitted between cells and promote amyloid aggregation in healthy cells. So that statement is not true.

The statement is now removed.

Furthermore, as I commented in the previous report, given the controversial claim that amyloidosis is due exclusively to loss of protein function, I think the authors should provide a small literature review of the state of the art to support their claim. However, they hardly provide any bibliography to support it, and if one looks for almost everything that can be found contradicts precisely their proposal. In fact, the physiological function of some amyloidogenic proteins, such as alpha-synuclein (whose aggregation in the form of Lewy bodies is the hallmark of Parkinson's disease), is not even well understood. So it is difficult to argue that it is their loss of function that gives rise to amyloidosis and not the toxic effect of any of the species involved in the amyloid cascade. I repeat that I would find this review very interesting, providing the state of the art in this direction, if a bibliographical study were to be carried out to support this hypothesis, instead of stating it almost without support.

7. Point 16 went deeper into the problem that had already been advanced in point 15 of the previous report, namely the lack of bibliography to support the authors' hypothesis. The authors' correction has been to add one line and three references, which is far from what would be considered necessary to accept their proposal with a certain degree of rigour. The authors propose a controversial hypothesis for the whole of amyloidosis and all possible amyloid fibrils, and provide hardly any literature and only for amyloid beta peptide. For the rest, I refer to what was said in the previous point. A hypothesis that goes against most of the experimental studies that have been carried out to date is a very interesting topic but one that requires a deep and well-argued discussion and with a multitude of studies to support it. This paper affirms this hypothesis without either of these two requirements.

Overall the article is better, but there are still misconceptions regarding the role of seeds in amyloid aggregation. Once those errors in the seeds role are corrected, the paper would be fine for the introductory part. Another thing is the issue of the role of loss of protein function in the origin of amyloidosis. I think that the authors have the opportunity to make a very interesting review if they develop it well, but now it is not. It is not a question that they have made any error that can be corrected, as has happened with the initiation of amyloid aggregation, but that they have hardly provided data to support their hypothesis. I think that the study they have done is far from being enough since they have focused on Alzheimer's disease without giving more examples, despite the fact that they propose a general mechanism. And within Alzheimer's disease they are based more on the lack of clinical studies than on actual studies that support their hypothesis. I am convinced that an excellent and interesting review can be made if the authors go deeper into this point.

We thank the reviewer for her/his suggestion. However, in this manuscript, which is aimed for a special issue on the biophysics of amyloid aggregation, we chose to focus on the structural and thermodynamic aspects of the amyloid phenomenon. We have recently published other reviews that focus more on the loss-of-function aspect of pathogenesis, which we now refer the reader to for more in-depth discussion and more references.

Round 3

Reviewer 3 Report

I believe that the authors have made the pertinent improvements and the paper is ready for publication in its current state.